# Gluten Free Wheat: Are We There?

**DOI:** 10.3390/nu11030487

**Published:** 2019-02-26

**Authors:** María Dolores García-Molina, María José Giménez, Susana Sánchez-León, Francisco Barro

**Affiliations:** 1Department of Plant Breeding, Institute for Sustainable Agriculture (IAS-CSIC), 14004 Córdoba, Spain; lolagmolina@hotmail.com (M.D.G.-M.); mjga06@ias.csic.es (M.J.G.); ssanchez@ias.csic.es (S.S.-L.); 2DAFNE Department, University of Tuscia, 01100 Viterbo, Italy

**Keywords:** gluten, coeliac disease, NCWS, transgenic wheat, RNAi, CRISPR/Cas9

## Abstract

Gluten proteins, major determinants of the bread-making quality of wheat, are related to several digestive disorders. Advances in plant genetic breeding have allowed the production of wheat lines with very low gliadin content through the use of RNAi and gene editing technologies. In this review, we carried out a comprehensive study of the application of these cutting-edge technologies towards the development of wheat lines devoid of immunogenic gluten, and their genetic, nutritional and clinical characterization. One line, named E82, showed outstanding nutritional properties, with very low immunogenic gluten and a low stimulation capacity of T-cells from celiac patients. Moreover, a clinical trial with non-celiac wheat sensitivity (NCWS) patients showed that the consumption of bread made with this E82 low gliadin line induced positive changes in the gut microbiota composition.

## 1. Wheat and Wheat Proteins

Wheat is one of the most important cereals in the world. Although starch is the major component of wheat grains (60–75%), the proteins of the grain (9–18%) are essential for bread-making quality. According to their functionality, wheat grain proteins are divided into two types: gluten and non-gluten proteins. About twenty per cent of the total grain protein corresponds to non-gluten proteins, comprising albumins and globulins, which have metabolic and structural functions with a minor role in wheat quality. In contrast, gluten proteins represent about 80% of the total grain proteins and they are mainly responsible for the rheological properties of the dough. These gluten proteins, also called prolamins given their high content of the amino acids proline and glutamine [1,2], include gliadins (α/β-, ω-, and γ-gliadins) and glutenins, comprising high molecular weight glutenin subunits (HMW-GS) and low molecular weight glutenin subunits (LMW-GS) (Figure 1). Most of the gliadins are monomeric proteins and they form intramolecular disulphide bonds; however, glutenins are polymeric complexes linked by inter- and intramolecular disulphide bonds to glutenins and gliadins. Both gliadins and glutenins form a viscoelastic network that traps the CO_2_ released during fermentation, providing the typical texture characteristics of the wheat bread. In this process, gliadins are responsible for the viscosity and extensibility of the dough. Nevertheless, glutenins provide elasticity and strength to the dough, contributing, especially the HMW-GS, to the formation of long polymers [3]. Gluten proteins are coded by multiple genes at complex loci present on chromosomes 1 and 6. In particular, α-gliadins are coded by genes at the *Gli-2* loci present on the short arm of chromosome group 6 [4], ω-, and γ-gliadins are genetically linked and are coded by genes at the *Gli-1* loci on the short arm of chromosome group 1, typical LMW-GS are coded by genes at *Glu-3* loci, genetically linked to *Gli-1* loci, and finally, HMW-GS are coded by genes at *Glu-1* loci present on the long arm of chromosome group 1 (Figure 2). The presence of gliadins and glutenins, and the balance between these two types of proteins is essential for the quality of the final product [5].

## 2. Wheat Pathologies and Gluten-Free Diet (GFD)

Wheat is associated with pathologies such as coeliac disease (CD), which affects about 1% of the population worldwide [6], non-coeliac wheat sensitivity (NCWS) [7] and allergies; baker’s asthma [8] and wheat-dependent exercise-induced anaphylaxis (WDEIA) [9].

Baker’s asthma is a respiratory allergy triggered by a wide range of wheat proteins that react with immunoglobulin E (IgE). Wheat proteins responsible for baker´s asthma comprise gliadins, glutenins, serine proteinase inhibitors (serpins), thioredoxin, agglutinin and several enzymes [10]. The α-amylase inhibitors are included among these enzymes, which are proteins soluble in chloroform: methanol (CM-like proteins) [8]. These α-amylase inhibitors have been described as the major group of proteins responsible for this allergy. Wheat is also responsible for WDEIA, which is an allergic reaction caused by combining the ingestion of wheat food and subsequent physical exercise. The major allergens associated with WDEIA are the ω-5 gliadins [9,11]. Palosuo et al. [12] suggested that the activation of transglutaminase in the intestinal mucosa could be provoked by the development of large allergen complexes responsible for triggering anaphylactic reactions during physical exercise in patients with WDEIA.

NCWS is a widespread and heterogeneous pathology. The disease has been described as a reaction to gluten proteins, in which allergic and autoimmune mechanisms have been excluded. In fact, other proteins such as metabolic proteins called α-amylase/trypsin inhibitors (ATI) [13], and FODMAPS (fermentable oligo-saccharides, disaccharides, monosaccharides and polyols) seem likely candidates to cause this pathology. Removing gluten from the diet is the only way to normalise the small-bowel mucosa as well as improving the symptoms. On the other hand, there are conflicting results about the existence of a wheat/gluten-induced inflammation in the majority of patients, as the mucosa from patients with gluten/wheat sensitivity does not express markers of inflammation, and their basophils are not activated by gliadin [14].

Coeliac disease is the most studied of these pathologies. It is an autoimmune disorder that occurs in genetically predisposed individuals triggered by gluten proteins from wheat (gliadins and glutenins), rye (secalins), barley (hordeins), oats (avenins), and also, all hybrids in which any of the toxic cereals are involved. CD has a strong environmental component, gluten, but also a genetic component, concerning the human leukocyte antigen (HLA)-DQ2 and HLA-DQ8 [15].

Gluten proteins are characterised by a high content of proline and glutamine residues making their complete digestion difficult. This gluten composition produces large peptides with immunostimulatory activity in the intestinal lumen [16]. These immunogenic peptides cross the intestinal epithelium and are deamidated by the tissue transglutaminase 2 (tTG2) in the lamina propria [17], providing a negative charge to gliadin peptides and hence enhancing their affinity to bind HLA-DQ2/8. It causes the activation of CD4^+^, triggering intestinal damage and malabsorption symptoms. For immunological reasons, 95% of coeliac patients present the HLA-DQ2 antigen and 5% contain the HLA-DQ8 [18]. The α-gliadin protein family is highly stimulatory because the 33-mer, the main immunodominant peptide for coeliac patients, is located in the repetitive region of these proteins [19]. The 33-mer peptide is resistant to gastric and pancreatic digestion, playing an essential role in the development of coeliac disease [16,20]. α-Gliadins also contain an additional DQ2-restricted epitope which partially overlaps with the 33-mer peptide [21]. However, it has been seen that not only the α-gliadins fraction is responsible for the stimulation of T-cells in people with CD. In fact, for the purpose of investigating the recognition profile of gluten immunogenic peptides in HLA-DQ2 coeliac patients, Camarca et al. [22] analysed the ability of stimulatory peptides to stimulate T-cell clones from CD patients. They concluded that there is indeed a substantial heterogenicity in the response of T-cells to gluten, also highlighting the stimulatory relevance of ω-, and γ- peptides.

In addition to the role played by gluten in the activation of gluten-specific T cells, it has been shown to affect the innate immune system. These “innate peptides” are not recognised by CD4^+^ T cells but induce an innate-like response in the epithelium and antigen-presenting cells (APCs). In this context, the α-gliadin p31-43 is the most studied peptide. Maiuri et al. [23,24] described this peptide as an innate immune response inducer, necessary to initiate the T-cell adaptive response. Moreover, the expression of both interleukin-15 (IL-15) and non-classical major histocompatibility complex (MHC) molecules, such as the MHC class I chain (MIC) and HLA-E, during the innate response, is mediated by the 33-mer peptide [24]. These molecules were shown to play an important role in intraepithelial (IEL)-mediated epithelial cell killing [25,26].

The only treatment for coeliac patients and all the other wheat pathologies is a gluten-free diet (GFD) for life. However, a GFD is difficult to follow because gluten is an additive widely used in the food industry, appearing in products which originally do not contain gluten such as meat, fish and many other foodstuffs. Because of this, the risk of transgressions in coeliac patients adhering to a gluten-free diet increases between 32% and 55% [27,28]. In addition, nutritionally gluten-free products could be less healthy because they are made with high amounts of fat and sugar to achieve a texture resembling the typical and unique wheat viscoelastic properties.

## 3. Towards Obtaining Wheat Lines with Low Immunogenic Peptides

The development of wheat varieties with reduced immunogenic gluten proteins could be noteworthy for CD and NCWS people, not only to improve their quality of life but also to reduce the incidence of those pathologies, given that a relationship between the amount and exposition to gluten and these diseases has been suggested [29]. The appeal of these wheat varieties could undoubtedly also apply to the general population, in particular for those who want to reduce the intake of gluten. The search for wheat varieties, which naturally are devoid of coeliac epitopes in the gliadin sequences encoded by the A, B, and D genomes of wheat, has been discussed for a long time. Hence, van den Broeck el al. [30], using monoclonal antibodies, analysed different wheat lines from Chinese Spring with partial deletions in the short and long arms of the group 6 chromosomes. They obtained a reduced T-cell stimulatory response in those lines that carried the deletion of α-gliadins in the D genome. These findings supported the study of van Herpen et al. [31] who showed that epitopes are not randomly distributed in the wheat genome and the three genomes contribute differently to the total epitope content and immunogenicity. Also, Spaenij-Dekking et al. [32] reported the existence of wheat genotypes with a low content of T-cell stimulatory sequences, suggesting that selection of less toxic lines through classical breeding could be possible. On the other hand, Juhász et al. [33] combined the use of both the genome sequence of the cultivar Chinese Spring and wheat protein/peptide databases. They carried out a comprehensive analysis of wheat genes that encode proteins related to digestive disorders and allergies, and their chromosomal locations. This reference mapping of immunostimulatory wheat proteins provides a new tool to select programs targeting traits, such as producing low-gliadin lines. With the exception of this last study, the rest of them described above were based on α-gliadin fractions. Thus, it cannot be concluded that they are exempt from other sequences that can stimulate the autoimmune response in coeliac individuals since it is known that many epitopes are present in other gliadin fractions [34], especially the ω-gliadins that could also play an important role in CD [19]. Moreover, the sequences of the individual genes within the same family of gliadins are very akin, and they might contain multiple and different T-cell stimulatory epitopes. This high complexity coupled with the fact that gliadin genes are inherited in blocks makes obtaining wheat lines with reduced toxic epitopes by using conventional breeding really difficult.

One promising approach would be to reduce the amount of CD immunogenic gluten proteins by the application of the latest developments in genetic engineering to toxic cereals. In this case, the silencing of gliadin genes, specifically the immunodominant α-gliadin fractions [35] could be achieved.

Given that gliadins are formed by groups of proteins encoded by multigene families that contain epitopes related to CD, the objective of several researchers has been the silencing of groups of gliadin genes rather than single genes [36,37]. In a first attempt, Gil-Humanes et al. [38] carried out the silencing of the γ-gliadins fraction using a D-Hordein promoter [39]. These transgenic lines were obtained from the *T. aestivum cv* Bobwhite (denoted as BW208, and here used as a control). In this work, they demonstrated that using RNAi it was possible to silence a complex gene family. Furthermore, the results obtained from A-PAGE gels and matrix-assisted laser desorption/ionization time-of-flight mass spectrometry (MALDI-TOF MS) revealed an increment in the other gliadin fractions, suggesting a compensatory effect as a result of silencing of γ-gliadins. As result, total gluten content, detected by monoclonal antibodies did not decrease but increased for some lines, indicating that a compensation mechanism with other gliadin fractions is operating, and the silencing of a single gliadin family does not provide wheat lines devoid of toxicity [40].

In the search for more comprehensive gliadin silencing, wheat lines were obtained with all three gliadin families silenced [40]. To achieve this, they used silencing fragments from the most conserved regions from α- and ω-gliadins, combined with different endosperm-specific promoters: the D-Hordein promoter (pDhp_ω/α) and the γ-gliadin promoter (pGhp_ω/α) [41]. In this way, more promising results were achieved, and competitive ELISA based on the R5 monoclonal antibody provided a significant decrease in the gluten content, obtaining the greatest decrease in the line named E82, with a percentage reduction of 98.1% in comparison with the control BW208 line [40]. Moreover, the success in obtaining low-gliadin lines was also corroborated by acid-polyacrylamide gel electrophoresis (A-PAGE), showing an effective down-regulation of gliadins from all three fractions. HPLC data, also obtained from those lines, showed a reduction of gliadins ranged from 69.8% to 87.9% in the transgenic lines named D783 and D793 respectively. There were no changes in the total protein content [40], which was explained by a compensatory increment of the non-gluten proteins and the HMW-GS fraction [42]. Moreover, further proteomic studies carried out using D783 and D793 lines, confirmed that gliadins and LMW-GS decreased in both transgenic lines. The decrease in the content of these proteins was compensated, specifically, by albumins and globulins, i.e., serpins, α-amylase/trypsin inhibitor family, β-amylase, all belonging to non-gluten proteins [43], and also by triticins [42], a lysine-rich globulin protein, whose increment could also explain the higher lysine content detected in those low-gliadin transgenic lines [44].

Grain protein profile and specific reduction of the different gliadin fractions can be achieved by using different silencing fragments for targeting prolamins in bread wheat [45]. Lines were analysed by proteomic and genetic techniques, and specifically, the gluten immunogenicity was determined by using competitive anti-gliadin 33-mer moAb. The results indicated a decrease in the gluten content in all silencing-fragment combinations except for lines with only γ-gliadins silenced (pghpg8.1 construction). Furthermore, the CD epitope analysis by mass spectrometry of pepsin and trypsin digested protein extracts, provided the identification of the E82 line as containing the lowest amount of CD immunogenic epitopes.

## 4. Stability of the Gliadin Silencing

The stability of gliadin silencing is a key factor for the usefulness of the low-gliadin lines, either for its introgression to commercial cultivars or as raw material for food processing. In order to quantify the robustness of this silencing, we have collected the data from several years of experiments with 15 different RNAi lines, derived from two wild type parental lines, carrying the hairpin ω/α fragment [45], and also from three nitrogen fertilization levels. Therefore, the data provide valuable information on the performance of the construct in different environments (Figure 3 and Figure 4). These data comprise values for several variables concerning the grain protein content as quantified by RP-HPLC: gliadin content and its fractions (omega-, alpha-, and gamma-gliadins); glutenin content and its fractions (high molecular weight- and low molecular weight-glutenin); the content of total gliadin and glutenin; and the gliadin to glutenin ratio. Values for the gluten content (parts per million; ppm) as determined by monoclonal antibodies are also available in most of the works. All data have been transformed into proportions to their respective wild type so that they are comparable. The objective is to eliminate or, at least, reduce the content in this toxic fraction of the gluten.

Overall, in the set of six trials, reductions of more than 75% in the gliadin content have been obtained with respect to the wild type, and in some cases, around 90% (Figure 3), of the value of the wild type line from which they are derived (see wild-type values in Table 1). However, the decrease is not of the same magnitude for all the gliadin fractions, producing a greater reduction, especially, in γ-gliadins (Figure 3). Of all the silenced lines, three of them were highly promising and therefore they were present in most of the published trials (D783, D793 and E82) [40,45,46,47,48]. Of these, lines D793 and E82 show a lower content of α- and γ-gliadins, which results in a reduction of 80% in the content of total gliadins (Figure 4). The E82 line presents, in addition, a significant reduction in the content of LMW glutenins [40] and, consequently, in the content of total glutenins (Figure 4). Comparing the variation in the prolamin content of these genotypes, it can be concluded that line E82 is of great interest in obtaining lines with steadily reduced toxicity.

The maximum allowed content of gluten in a food to be considered gluten-free by Codex Alimentarius (CODEX STAN 118-1979), European Commission Regulation (EC 41/2009) and the U.S. Food and Drug Administration (FRDoc 2013-18813) is 20 mg/kg (20 ppm). The method recommended by CODEX for gluten quantification is based on the monoclonal antibody (MoAb) R5 [49]. The lines with gliadins strongly down-regulated by RNAi have an average reduction of 90% of the gluten content (Figure 3), being even higher for D793 and, especially for the E82 line (more than 95%, Figure 4). If we consider that the average value of the gluten content of the BW208 wild type flour is more than 100,000 mg/kg (Table 1), this represents a huge reduction in gluten content and therefore in toxicity. However, the gluten content of E82 flour is still higher than the limit to be considered gluten-free. In any case, this flour could be used as raw material for mixing with other flours, especially gluten-free, to obtain further reductions in the gluten content. However, it is important to note that further reductions in gluten content can be obtained after fermentation and cooking and, for this E82 line, the gluten content of the final bread is around one-third of the flour [50].

## 5. Are These Lines Closer to the Table?

Low-gliadin wheat lines are being developed as an alternative cereal with a better nutritional profile and organoleptic properties, and improved bread making than that currently used for gluten-free products. Different groups of patients/persons may benefit in some way from these properties of low-gliadin lines: (i) people with coeliac disease; (ii) people suffering from non-coeliac wheat sensitivity (NCWS), although gluten seems not to play an essential role in this pathology [51], (iii) people genetically predisposed to develop coeliac disease, but not showing symptoms, in order to reduce the amount and exposition to gluten; and (iv) people who despite not having any gluten-related pathology have decided to reduce the intake of gluten in their diet.

With the aim to ensure the traceability of the low-gliadin lines, avoiding possible cross-contamination, a robust classification model to discriminate between low-gliadin and standard gliadin wheat lines was developed using near infrared spectroscopy (NIRS) technology [52]. In this study, a total of 535 samples of grain and 570 samples of flour obtained during 5 years were analysed. The results showed better discrimination using flour than grain because the 99% of the flour samples and the 96% of the grain samples used during the validation were correctly classified. These results were quite promising from the point of view of both the farmer and the industry, because, using this technology, one could measure the traceability of gluten-free or low-gluten products made with these lines with very low gliadin content [52].

On the other hand, the requirement of fertilisation, especially nitrogen, for low-gliadin wheat is totally unknown, but it is of concern for the farmers as it is well known that protein accumulation during grain filling is strongly influenced by nitrogen fertilisation [53] and it can produce modification on the grain protein composition and yield [54,55]. Therefore, fertilisation strategies for these lines are undoubtedly important in order to keep gluten levels as low as possible with no compromise of yield and plant growth. Different fertilisation strategies and their impact on the silencing of gliadins and the accumulation of other grain proteins were studied in low-gliadin wheat lines [47]. They carried out two fertilisation strategies; in the first, three nitrogen fertiliser rates (120, 360 and 1080 mg N) applied at the sowing stage were studied; in the second, two nitrogen fertiliser rates (120 and 1080 mg N) added at different stages according to the greatest demand by plants were evaluated. The results showed that split nitrogen and adding 120 mg N obtained a reduction in gluten content (ppm). Moreover, of all the lines analysed in this study, the line E82 was the only one in which nitrogen increments did not result in increased gluten content (ppm), meaning that increasing fertilisation for this line does not provide higher gluten content.

Focusing on the end use of these lines, would it be possible to produce acceptable bread with good organoleptic properties using the flour of these low-gliadin lines? Gil-Humanes et al. [44] studied the organoleptic, nutritional, and immunotoxic properties of loaves of bread made with flour from different low-gliadin lines, normal wheat bread containing gluten, and bread made using rice flour as this is an important ingredient in gluten-free foods. The comparison among them showed that bread made with the low-gliadin flour presented sensory and baking properties similar to those made with normal bread containing gluten. The low-gliadin bread also had better organoleptic properties and acceptance than rice bread. Regarding the nutritional value, flour from these low-gliadin lines contained higher lysine content than that of normal flour containing gluten. Lysine is an essential amino acid that cannot be synthesised in animals it must be supplied through the diet, and its content is low in standard wheat. Specifically, bread made using the line E82 showed the best sensory and nutritional properties as well as the strongest gluten reduction in loaves (~96% reduction) [44].

Thus, the line E82 has been outstanding given its low content of CD toxic epitopes [40,44] as well as no increase in toxicity with different levels of fertilization [43]. Also, we have seen its good baking characteristics and its high content of lysine improving its nutritional profile [42]. The safety of this low-gliadin line was evaluated using Sprague Dawley rats [48]. Animals were fed with whole wheat flour from the line E82 using three increasing doses for 90 days. Some parameters analysed were the function of the liver and kidney, haematological and metabolic blood, organ weights, etc. No significant differences were found between rats fed with flour from the line E82 and those with the wild-type wheat, concluding that line E82 did not trigger any adverse effects in the health of the rats.

All these characteristics made the E82 line the best candidate to carry out clinical trials with NCWS and/or coeliac volunteer patients. Eleven patients with non-celiac gluten sensitivity (NCGS) were included in a trial to evaluate the symptoms, acceptance, and digestibility of bread made from the low-gliadin E82 line, in comparison to gluten-free bread [50]. All participants were on a gluten-free diet for at least 6 months prior to the trial. The trial was organised in two phases, with a duration of seven days each. During the basal phase, patients were on a gluten-free diet and consuming 100–150 g of a gluten-free bread of their choice daily. In the second phase, patients were on a gluten-free diet but consuming 100–150 g of a bread made with the E82 low-gliadin line. After each phase, clinical questionnaires based on the Gastrointestinal Symptom Rating Scale (GSRS), and sensory questionnaires about the acceptability of the bread were completed, and stool samples collected for gluten immunogenic peptide (GIP) determination and gut microbial DNA analysis. Results were highly promising as no significant differences in the GSRS questionnaires were found between the two phases, i.e., no changes in symptoms between the two phases. Five patients showed undetectable GIPs content in the E82 phase. However, one interesting finding is that the intake of the E82 bread provided a microbial profile which has a positive impact on gut permeability, and better functioning of the intestinal barrier. Specifically, the most efficient butyrate-producing bacteria, such as the *Roseburia* and *Faecalibacterium* genera (*F. prausnitzii*), were promoted during the consumption of bread made with E82 line [50].

## 6. Future Prospects

Low-gliadin lines fall under genetically modified organisms (GMO) regulation as they were produced by RNAi technology. One important issue is how this kind of product could be, or not be, accepted by the general population or by those suffering any of the gluten-related intolerances. Although GM crops have been internationally produced since the first GMO hit grocery stores in 1994 [56], and there are many peer-reviewed studies on their economic, environmental and health benefits [57], many countries still have concerns about GMOs adoptions. Opposition to GMO by ecological organizations is very strong in Europe, but with worldwide impact. Unlike other breeding methods, the application of genetic transformation is highly and strictly regulated in the European Union (EU). This is completely inconsistent with the fact that cultivation of GMOs is practically prohibited in the EU but importation is allowed [58]. As a consequence of this strict regulation, the general population have concerns about GMOs on several levels, including their environmental effects or whether GM foodstuff could have any risk for consumption. After many long-term multigenerational studies on the potential effects of diets containing GMOs, there is no evidence of any health hazards on animal feed, suggesting GM plants are nutritionally equivalent to their non-GM counterparts and can be safely used in food and feed [59]. Despite these studies, consumer acceptance is still conditioned by the risk perception they have of introducing food into their consumption habits developed through technologies that are being biasedly demonised by ecological organizations. For these reasons, considerable effort needs to be directed towards understanding people’s attitudes and to clearly explain the benefits of this gene technology to them. In the case of the low-gliadin RNAi lines, such as E82, the benefits are clear, at least for people who suffer any of the pathologies related to gluten; it is wheat that does not contain immunogenic proteins. Furthermore, any potential risk could be recognised and evaluated before public consumption to show that the product does not trigger any other adverse effects.

On the other hand, emerging targeted genome editing technologies represent a new powerful tool for plant breeding. Site specific nucleases (SSN) can be programmed to cut or modify DNA sequences, resulting in targeted mutation or in the correction of specific sequences. SSN, like CRISPR/Cas9, typically produce double-stranded breaks (DSB) that are repaired by non-homologous end-joining (NHEJ) or the homologous recombination (HR), depending on whether undamaged homologous DNA is present to act as a repair template [60]. In contrast to transgenic modifications that involve the introduction of foreign DNA sequences into a genome, gene editing can generate genetic variation through precise and direct changes in genes of interest with no integration of foreign DNA or, if so, null segregants, containing no recombinant DNA but retaining the desired mutations, can be easily recovered. Such edited plants might not fall into the GMOs definition and could be considered as non-transgenic plants.

In terms of providing gluten-free wheat, CRISPR/Cas9 can be used to eliminate or reduce the content in the toxic fraction of the gluten. As Sanchez et al. [61] showed, high mutation frequency using CRISPR/Cas9 of the α-gliadin gene family in bread and durum wheat was obtained, providing lines with up to 85% reduced immunoreactivity compared to that of the wild type. These low-gluten transgene-free wheat lines constitute the first step in the development of wheat lines totally devoid of gliadin genes. The main challenge from now is to take advantage of the full potential of the CRISPR/Cas9 technology to precisely modify gliadin genes, suppressing their immunogenic capacity but keeping their functionality and organoleptic characteristics. Base editors, which are composed of a deaminase domain and a Cas9 variant (D10A nickase (nCas9) or catalytically deficient Cas9 (dCas9)) can be used to specifically introduce mutations in CD epitopes, changing the amino acid composition of such epitopes, and breaking their recognition by antigen presenting cells. The efficiency of these base editors have been greatly improved, and the use of different Cas proteins, like Cpf1, also expands the application of this technology to AT-rich regions [62].

## 7. Conclusions

RNAi technology was used to produce a set of wheat lines with a reduced content of gliadins. Among those lines, E82 is outstanding because of its low capacity to elicit an immunogenic response, its organoleptic properties and agronomic performance. In addition, the consumption of bread made with this low-gliadin line promotes a better gut microbiota profile than gluten-free bread in NCWS patients. The line E82 has shown to be an exceptional candidate as an alternative for the preparation of products with reduced gluten content to improve the organoleptic and nutritional profile of the diet of NWGS sufferers. The new breeding techniques, like CRISPR/Cas9, will contribute to having gluten-free wheat on hand on our tables.

## Figures and Tables

**Figure 1 nutrients-11-00487-f001:**
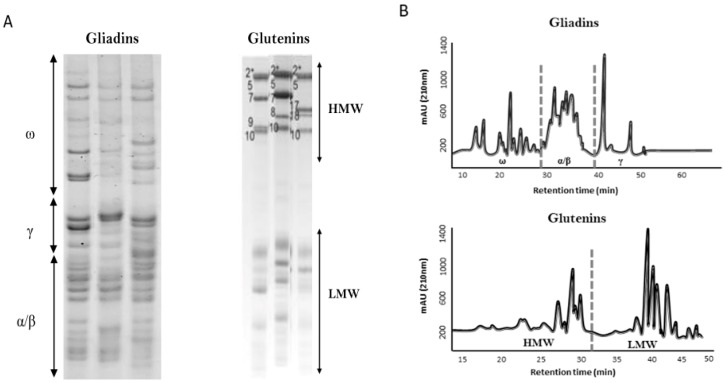
Gliadins and glutenins fractions revealed by using acid polyacrylamide gel electrophoresis (A-PAGE) and sodium dodecyl sulfate polyacrylamide gel electrophoresis (SDS-PAGE), respectively (**A**) and RP-HPLC (**B**). ω, ω-gliadins fraction; γ, γ-gliadins fraction; α/β, α/β-gliadins fraction; HMW, high molecular weight glutenins subunit; LMW, low molecular weight glutenins subunit.

**Figure 2 nutrients-11-00487-f002:**
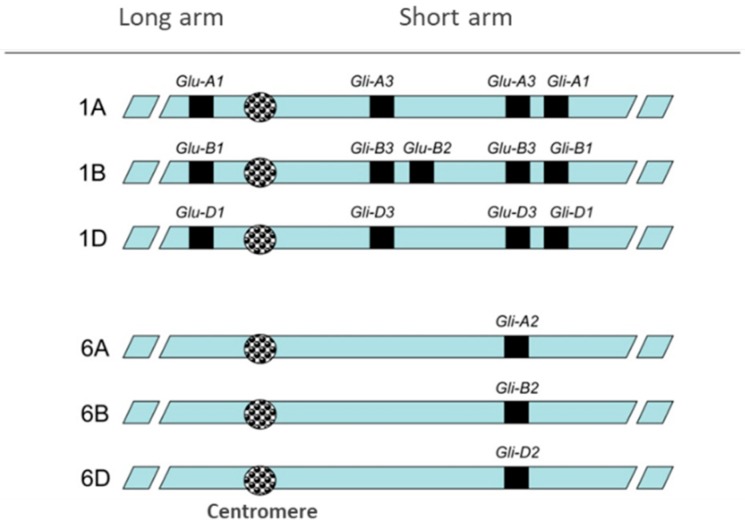
Gliadins and glutenins genes location in wheat chromosomes 1 and 6. Reprinted from thesis manuscript of Gil-Humanes (https://helvia.uco.es/handle/10396/5233).

**Figure 3 nutrients-11-00487-f003:**
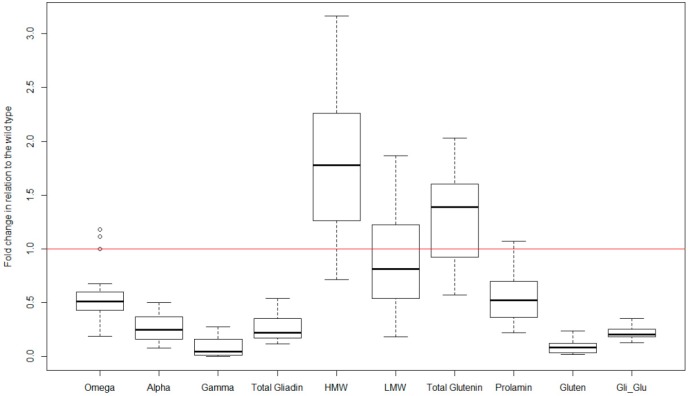
Boxplot of the low-gliadin transgenic lines included in the set of trial published to date. Variables: content of omega-, alpha- and gamma-gliadin, total gliadin content, HMW- and LMW-gluten, total glutenin content, prolamin content (gliadin + glutenin contents), gluten (as measured by monoclonal antibody R5), and gliadin to glutenin ratio. Data have been transformed to fold change of their respective wild-type genotype.

**Figure 4 nutrients-11-00487-f004:**
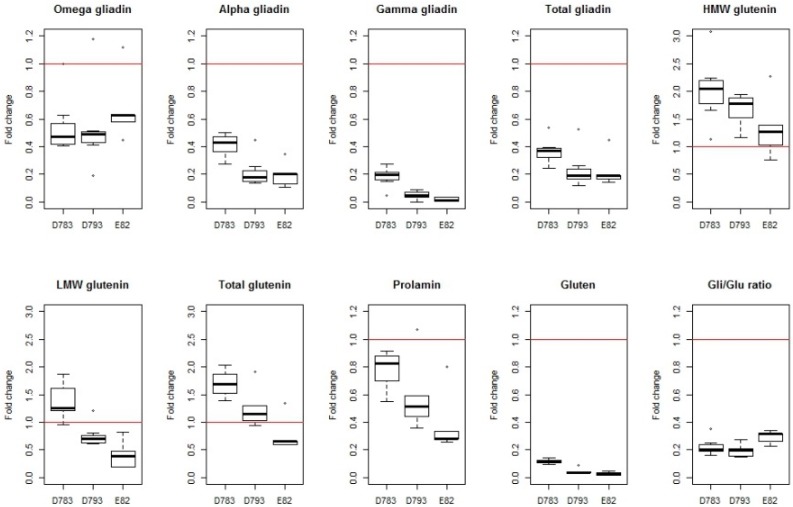
Boxplot of fold change of variables for lines D783, D793 and E82 in the set of trials published to date; Prolamin and prolamin fractions contents, gluten content and Gli to Glu ratio. Raw data have been collected from Gil-Humanes et al, 2010 [40]; Barro et al, 2016 [45]; Pistón et al, 2013 [46]; García-Molina et al, 2017 [47]; Ozuna & Barro, 2017 [48]. HMW, high molecular weight; LMW, low molecular weight.

**Table 1 nutrients-11-00487-t001:** Content of prolamin and prolamin fractions for wild-type line BW208; gluten content as measured by R5 monoclonal antibody, and gliadin to glutenin ratio.

Variable	Min.	1st Qu.	Median	Mean	3rd Qu.	Max.
Omega (mg/g)	7.7	10.8	14.1	14.9	19.5	21.7
Alpha (mg/g)	14.9	27.5	37.7	33.3	39.1	42.6
Gamma (mg/g)	8.9	25.5	26.7	24.5	27.8	28.8
Total gliadin (mg/g)	31.5	65.0	77.8	72.4	83.5	93.1
HMW (mg/g)	6.3	7.1	12.4	12.3	17.0	19.4
LMW (mg/g)	12.6	13.0	18.6	18.5	21.9	28.1
Total glutenin (mg/g)	18.9	20.0	30.9	30.8	38.8	47.5
Prolamin (mg/g)	51.9	84.0	108.7	103.1	121.8	140.6
Gluten (mg/kg)	45,266	67,134	134,673	111,548	149,843	166,942
Gliadin to Glutenin ratio	1.6	2.0	2.2	2.5	3.4	3.5

HMW, high molecular weight, LMW, low molecular weight.

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
