# Peer review of "Gluten Free Wheat: Are We There?"

_nutrients, 2019, doi:10.3390/nu11030487_

Round 1
Reviewer 1 Report
The paper under review is produced by a group of researchers who have specific expertise in the field.
It is clear and accurate, enjoyable also by an average reader who wants to increase her/his knowledge in the properties and effects on one of the most frequent triggers of food intolerance.
I have only a major comment. In my opinion, a section is missing, that is the summary of the knowledge of the effects of gluten /wheat in diseases different from celiac disease. In particular, I suggest adding a paragraph on wheat allergy and another one on non-celiac wheat/gluten sensitivity. What is known so far?
Author Response
Note: Reviewer text extracts are reported in full (in bold). The symbol [A] will be used to identify the corresponding response in normal font.
I have only a major comment. In my opinion, a section is missing, that is the summary of the knowledge of the effects of gluten /wheat in diseases different from celiac disease. In particular, I suggest adding a paragraph on wheat allergy and another one on non-celiac wheat/gluten sensitivity. What is known so far?
[A] We completely agree with this suggestion and we have changed the section called “Coeliac disease and gluten-free diet (GFD) for the section “Wheat pathologies and gluten-free diet (GFD). This section comprises a summary of wheat pathologies suggested by the reviewer.
Reviewer 2 Report
This is an excellent review of a topic that is not well understood by clinicians and will contribute significantly to the literature.
I have only one minor comment to make -
In line 74 it is entirely reasonable to just use 'CD4' rather than 'T cell cluster of differentiation 4' which whilst strictly correct is confusing and the terminology is well accepted without the need to expand it.
The English does need a little editing to make it more easily comprehensible.
Author Response
Note: Reviewer text extracts are reported in full (in bold). The symbol [A] will be used to identify the corresponding response in normal font.
In line 74 it is entirely reasonable to just use 'CD4' rather than 'T cell cluster of differentiation 4' which whilst strictly correct is confusing and the terminology is well accepted without the need to expand it.
[A] We agree with the reviewer. We have removed “T cell cluster of differentiation 4” and we have used “CD4”.
The English does need a little editing to make it more easily comprehensible.
[A] The English has been improved.
Reviewer 3 Report
In the submitted manuscript, Garcia-Molina et al. highighted the potential as well as current limits of the gliadin gene silencing strategy, obtained by RNAi technology, specifically applied by this group to obtain wheat lines with reduced toxicity.The study is generally well designed, documented and presented in the manuscript with cognate literature. However, I think that this review still suffers from some limitations in the introductive paragraphs, which should be addressed:
Chapter 2 should contain more recent literature data, now describing a heterogeneity of HLA-DQ2/DQ8 restricted gliadin epitopes, not only from a-gliadin. This should also apply to a more defined role of gliadin molecules in innate immunity, as found by the Jabri’s team.
Chapter 3 should encompass new topical literature about the mapping of seed-borne immune-responsive proteins in wheat (Juhász et al., Sci. Adv. 2018; 4), based on the recently sequenced wheat genome.
Minor points:
Please correct English in the following sentences: page 1, lines 38-39; page 3 line 74; page 4, line 162 (different years of what?); line 171, the sentence was redundant.
Fig.3, please insert "fold change" on the y-axis.
Author Response
Note: Reviewer text extracts are reported in full (in bold). The symbol [A] will be used to identify the corresponding response in normal font.
Chapter 2 should contain more recent literature data, now describing a heterogeneity of HLA-DQ2/DQ8 restricted gliadin epitopes, not only from a-gliadin. This should also apply to a more defined role of gliadin molecules in innate immunity, as found by the Jabri’s team.
[A] We completely agree with the reviewer and recent literature data regarding to the heterogeneity of HLA-DQ2/DQ8 restricted gliadin epitopes has been included. Also, we have described the role of gliadin molecules in innate immunity.
Chapter 3 should encompass new topical literature about the mapping of seed-borne immune-responsive proteins in wheat (Juhász et al., Sci. Adv. 2018; 4), based on the recently sequenced wheat genome.
[a]We appreciate the suggestion of the reviewer and the proposed study about the mapping of seed-borne immune-responsive proteins in wheat by using the sequenced wheat genome has been included in our work.
Minor points:
Please correct English in the following sentences: page 1, lines 38-39; page 3 line 74; page 4, line 162 (different years of what?); line 171, the sentence was redundant.
[A]English has been corrected. The redundant sentence has been removed.
Fig.3, please insert “fold change” on the y-axis.
[A] “Fold change” has been inserted on the y-axis from the Fig.3.
Round 2
Reviewer 1 Report
line 87 the word ' wheat' should be replaced by 'gluten'. NCGS patients exclude all sources of gluten
I suggest to add a sentence about conflicting results about the existence of a wheat /gluten-induced inflammation in the majority of patients, who in placebo-controlled trials do not recognize gluten in ther diet
Author Response
Note: Reviewer text extracts are reported in full (in bold). The symbol [A] will be used to identify the corresponding response in normal font.
The word ' wheat' should be replaced by 'gluten'. NCGS patients exclude all sources of gluten.
[A] According to the reviewer's suggestion, we have decided to make the following changes:
Line 82, the word "wheat" has been removed
Line 85, the word "wheat" has been replaced by "gluten".
I suggest to add a sentence about conflicting results about the existence of a wheat /gluten-induced inflammation in the majority of patients, who in placebo-controlled trials do not recognize gluten in ther diet
[A] We agree with the reviewer and the following paragraph has been introduced "On the other hand, there are conflicting results about the existence of a wheat / gluten-induced inflammation in the majority of patients, as the mucosa from patients with gluten / wheat sensitivity does not express markers of inflammation, and their basophils are not activated by gliadin " (Lines 87-90).